# Retention rate of vaccination card and its associated factors among vaccinated children aged 12—23 months in Ethiopia: Multilevel logistic regression analysis

**Abiyu Abadi Tareke**[1]*, **Atikaw Tewabe Ayelign**[2], **Thomas Kidanemariam Yewodiaw**[3], **Enyew Woretaw Shiferaw**[4], **Habitu Birhan Eshetu**[5], **Ermias Bekele Enyew**[6]

**1** Amref Health Africa in Ethiopia, COVID-19 Vaccine/EPI Technical Assistant at West Gondar Zonal Health Department, Gondar, Ethiopia, **2** West Gondar Zonal Health Department EPI Officer, Gondar, Ethiopia, **3** Department of Child Health Janamora Woreda Health Office, Gondar, Ethiopia, **4** Department of Maternal and Child Health (MCH), West Gondar Zonal Health Department, Gondar, Ethiopia, **5** Department of Health Education and Behavioral Sciences, Institute of Public Health, College of Medicine and Health Sciences, University of Gondar, Gondar, Ethiopia, **6** Department of Health Informatics, School of Public Health, College of Medicine and Health Sciences, Wollo University, Dessie, Ethiopia

* abiyu20010@gmail.com

**Data Availability Statement:** All relevant data for this study are within the paper.

## Abstract

### Background

Vaccine card is a crucial tool for gauging vaccine coverage. It is imperative to hold these health cards to have well-fitted data which are crucial in reaching data-driven decisions in the era of immunization surveillance and monitoring processes. However, there is limited knowledge about the retention rate of vaccination card and its associated factors in Ethiopia.

### Objective

This research aimed to assess the retention rate and associated factors of vaccination card in Ethiopia, using data from the 2016 Ethiopian demographic health survey.

### Methods

This study included a total of 1304 (weighted) children aged 12—23 months who were vaccinated and provided with a vaccination card. We used a multilevel logistic regression model to analyze factors associated with vaccination card retention. We considered factors to be statistically significant if they had a p-value of less than 0.05 with a respective 95% confidence interval.

### Result

Among the cohort of 1,304 immunized children, it was observed that 684, representing 52.5% (95% CI: 49.7%—55.2%), were able to present their respective vaccination card during the interview time. According to the results of the multilevel logistic analysis, there is a

**Funding:** The authors received no specific funding for this work.

**Competing interests:** The authors have declared that no competing interests exist.

**Abbreviations:** AIC, Akaike information criteria; AOR, : Adjusted Odds Ratio; VC, vaccination card; BIC, Bayesian information criteria; EDHS, Ethiopian Demographic health survey; DIC, Deviance Information Criterion; ICC, Intra-Class Correlation; LLR, Log-Likelihood Ratio; MOR, Median Odds Ratio, and; PCV, Proportion of Variance Change.

considerable reduction in the rate of vaccination card retention by 65% (adjusted OR 0.35, 95% CI: 0.19—0.65) and 37% (adjusted OR 0.63, 95% CI: 0.4—0.91) for individuals who are rural residents and those who are fully vaccinated, respectively. Furthermore, it is noteworthy to mention that individuals originating from socio-economic backgrounds with low poverty levels exhibit a 59% increase in vaccination card possession (adjusted OR 1.59, 95% CI: 1.11—2.50).

## Conclusion

This study revealed a low rate of holding vaccination cards. Place of residency, wealth status, and vaccination status were factors that contributed to the change in the vaccination card retention rate. It is advisable to customize the interventional strategy by taking into account the individual's residency, immunization status, and degree of poverty within the community, to achieve a favorable rate of holding vaccination cards.

## Introduction

The World Health Organization (WHO) initiated the Expanded Program of Immunization (EPI) back in 1974 intended to mitigate morbidity and mortality rate of under five children caused by vaccine-preventable diseases (VPD) globally [1, 2]. Globally, over two million fatalities are prevented annually through immunization. Vaccination cards (VC) are regarded as pertinent public health documents that encompass crucial details about vaccination dates, types of antigens, number of administered doses and other integrated health services [3]. Surveys based on population probability are often regarded as the "gold standard" for evaluating vaccination coverage, as they are free from certain biases and concerns [4]. In these surveys, vaccination status is commonly ascertained using household-retained vaccination cards, occasionally supplemented by parental recall [5].

Measuring vaccination coverage holds great importance for both national and local health programs, serving as a foundation for programmatic and policy decisions. It enables the evaluation of immunization services, thereby facilitating informed decision-making. Additionally, assessing the extent of immunization coverage offers valuable indications of whether significant advancements are being made in attaining vaccination objectives. To determine the extent of vaccination coverage within a particular region, the most relevant health card activity would be to ensure the preservation of VC.

Retaining vaccination cards holds significant importance for various stakeholders, including nations, communities, families, and individual children: Nationally vaccination cards aid national immunization programs by recording vaccination coverage, facilitating monitoring and evaluation, enhancing surveillance, and enabling quicker outbreak response. For the family, it enables adherence to vaccination schedules, ensuring timely protection, for the child, insuring children receive all recommended vaccines, promoting long-term health and well-being as well as helping healthcare providers in managing a child's healthcare, allowing informed decision-making [3].

Furthermore, it also serves as a source of information for care providers in terms of vaccine efficacy and adherence. Academic researchers: The information inscribed on VC can be of great advantage to researchers, as it can be used as a source of documentation while

conducting surveys, estimating vaccine timeliness, and determining the status of vaccination completeness.

The retention rate of VC refers to the percentage of individuals who still have their VC after receiving their vaccines. The possession of VC is viewed as a favorable factor in augmenting the percentage of children who have received complete vaccination (all the WHO-recommended antigens) [6–8]. Additionally, having health cards was found associated with favorable access to primary health care services [9].

In Ethiopia, the health institution-based documentation rate is found lower[10–13] and poor quality [10]. The rate of routine immunization vaccination card retention is not measured and the number of published articles concerning the VC retention rate of caregivers and associated factors in Ethiopia are limited. This study determines the rate of VC and factors contributing to the change in card retention among 1,304 vaccinated children.

## Methods and materials

### Study design and period

This study utilized data obtained from the fourth cross-sectional demographic health survey of Ethiopian communities, administered from January 18th to June 27th in the year 2016.

### Study setting

Ethiopia, the African nation under consideration, stands as the second-most populous country on the continent. As per the Population and Housing Census (PHC) projection of 2007, Ethiopia's total population was anticipated to approach nearly 110,000,000 by the year 2030. Based on data from the 2007 Census, it can be inferred that a substantial proportion of individuals were residing in rural areas, accounting for a significant majority (83.6%) of the population. The average family size was estimated to be 4.7 individuals per household. Additionally, it was observed that a considerable percentage (47%) of women belonging to the population cohort analyzed were between the ages of 15 and 49 years. [14].

According to the 2016 EDHS, the average number of children born to a woman over her lifetime (total fertility rate) was 4.6 children per woman [15]. However, the World Bank reported a fertility rate of 4.159 in Ethiopia for the year 2021 [16]. In 2019 Among children aged 12─23 months, 44% have received all basic vaccinations at some point, with 40% receiving them by the appropriate age [17].

### Population

**Source population.**   Children aged less than 2 years in Ethiopia.

**Study population.**   Children ranging from 12 to 23 months of age who have undergone vaccination for any antigen.

**Dependent variable.**   The dependent variable for the current research is the rate of retention of VC (yes/no) (computed as the percentage of children who have ever owned a VC and were capable of bringing it during the interview session).

**Independent variables.**   Based on our thorough review of literatures, we have identified maternal age, maternal education levels (ranging from none to primary, secondary and higher), the working status of the caregiver (employed or not), household size, media accessibility, the child's gender, vaccination status as individual level factors and community level poverty and literacy as community-level factors as the independent variables for this particular study.

**Operational definitions.** *Media Access.* This is determined by a combination of whether the respondent reads the newspaper, listens to the radio, and watches television. If the respondent has been exposed to at least one of these three media, it is labeled as "exposed" and coded as "1". On the other hand, if the respondent has not been exposed to any of the three media, it is coded as "not exposed" and given a code of "0".

*Immunization status.* A child is classified as "fully vaccinated" if they have received one dose of BCG vaccine, three doses each of polio vaccine, pentavalent vaccine (DTP-hepB-Hib), and pneumococcal conjugate vaccine (PCV), two doses of Rotavirus vaccine, and one dose of measles vaccine. Otherwise, if any of these criteria are not met, the child is considered "not fully vaccinated." [18].

*Community level of poverty.* community poverty level is determined based on the proportion of households assigned to the poorest and poorer wealth index categories. Households falling at the median value and above are classified as having a high poverty level, while those falling below the median value are categorized as having a low poverty level. The median is chosen as the cutoff point due to the skewed distribution of the variables. A similar categorization approach was applied to determine community-level educational status.

**Data collection tools and procedures.** The secondary data utilized in this study were obtained from the demographic health Survey Repository, which contains anonymized survey data collected by the Ministry of Health. The primary data were collected using standardized survey instruments, including household questionnaires administered by trained enumerators. These questionnaires covered various topics, including demographic information, health status, healthcare utilization, and other relevant indicators [19].

The survey employed a multistage stratified sampling design to ensure representation at the national, regional, and district levels. Household selection was conducted through systematic random sampling, with households selected based on probability proportional to size. Data collection was carried out through face-to-face interviews with household members, following established protocols to maintain consistency and quality throughout the process [19].

The website of DHS measure, http://www.dhsprogram.com/, was utilized to acquire registration for entry into the Ethiopian DHS Datasets necessary authorization was granted for the requested tools to be accessed. All necessary data were obtained from the website of the Demographic and Health Surveys Program, in accordance with the requirements.

**Data quality control.** The survey's data collectors documented that the questionnaires underwent a pre-testing phase in all three local languages (Amharic, Afaan Oromo, and Tigrigna) to ensure absolute clarity and understanding for the respondents [19]. Quality control measures were implemented throughout the data collection process to identify and address any inconsistencies, errors, or discrepancies in the collected data. These efforts aimed to uphold the integrity and reliability of the survey data and enhance the validity of the study findings.

**Sample size and sampling procedures.** A two-stage stratified clustered sampling procedure was implemented across all 11geographic administrative areas, which includes 9 regions and 2 city administrations. The initial stratum comprised 645 enumeration areas (EAs) that were chosen proportionally to the EAs size of the nine geographical regions and two administrative cities. In the second stratum, each of the eleven administrative divisions was subdivided into urban and rural residents, creating a total of 21 sampling strata. Using an equal probability technique, 28 households were selected from each cluster. From the 28 households, 1944 children aged 24─35 months were excluded from this study because of 12─23 months age group provides a snapshot of the current state of the country's performance regarding vaccination card utilization. This age group represents children who have recently completed or are in the process of completing their primary vaccination series, offering valuable insights into

vaccination practices and documentation during this critical period. The study ultimately involved a weighted sample of 1304 vaccinated alive children [19].

## Data analysis

The STATA/SE version 16.0 utilized to review, sort, recode, data analysis and statistical modeling, and the data. In order to ensure survey representativeness, and to acquire dependable statistical estimates, the data were weighted by applying the "svyset" command in STATA to account for the effects of the survey's complex sampling design or the hierarchical nature of the Ethiopian Demographic Health Survey (EDHS) dataset. This command was applied to each analysis conducted in this study.

**Multilevel regression model.** A multilevel logistic regression model was utilized to determine the factors associated with the retention of VC in Ethiopia. Multilevel analysis is particularly advantageous for examining data with nested structures like DHS data, as individual-level characteristics are more likely to be correlated within the same cluster or enumeration area than with those from other clusters. This can lead to a violation of the assumption of observation independence in standard regression models. Multilevel analysis is a valuable tool that can address the lack of independence of observations when analyzing nested or hierarchical data. Four distinct models were created to conduct this analysis. The null model, also known as the random intercepts model, was one of these models and does not include any child's or cluster's characteristics. It was used to determine the extent of cluster variability on VC retention levels. To assess the variability of the cluster, we computed the Intra-Class Coefficient (ICC), median odds ratio, and Proportional Change in Variance (PCV). The ICC gauges the percentage variation caused by community-level variables, whereas the PCV determines the proportional shift in the community-level variance between the null model and the subsequent models [20].

The Odds Ratio (OR) used by the MOR to describe area-level variance is the median value of the distribution of ORs obtained when two children with the same covariate values are selected from two different areas. These areas are compared with one another based on their respective VC retention levels, with one representing a higher level of retention than the other. In the absence of area-level variation, the Median odds ratio (MOR) equals 1. The presence of clustering and heterogeneity between areas in the outcomes of VC retention level was examined using intercept-only models (null model), which estimated the value of ICCs and MORs.

The fitness of each model was evaluated using various parameters such as the Likelihood Ratio test (LR), deviance, Akaike Information Criteria (AIC), and Bayesian Information Criteria (BIC). The model that displayed the lowest value among the four fitness parameters was chosen as the most suitable model. Factors were deemed statistically significant based on the adjusted odds ratio with 95% CI and p-value $<0.05$.

## Ethical consideration

This study employed a dataset comprised of demographic health surveys that are representative of the nation. Consequently, there is no need for ethical approval. However, the datasets utilized in this study were acquired through a process wherein a comprehensive explanation of the study's objectives and indispensability was presented.

The DHS dataset was procured through registration and request via the online database (www.dhsprogram.com), followed by receipt of an authorization letter that enabled download of the requested dataset.

# Result

The study population included 1,304 caregivers from across Ethiopia. Most mothers were 25─34 years old (54.74%) and had no formal education (56.54%), though about a third had completed primary school (32.17%). Approximately half of mothers worked outside the home (50.64%). Household media access was limited (60.50% had no access), and child vaccination status was mixed (46.81% partially, 48.28% fully vaccinated). The sample was predominantly rural (84.12%), with the largest proportion from Oromia region (37.76%). Levels of community poverty and literacy were relatively high (52.44% low poverty, 78.22% high literacy) (Table 1).

## Rate of VC retention

The retention rate of VC in Ethiopia is 52.5% (with a 95% confidence interval of 49.7% to 55.2%). A noteworthy observation was made concerning the retention rate of participants hailing from various regions of Ethiopia. Specifically, those residing in Addis Ababa exhibited a remarkably high retention rate of 91%, while those from Harari, Tigray, and Amhara displayed retention rates of 65%, 64%, and 61%, respectively. Moreover, residents of Dire Dawa also demonstrated a relatively high retention rate at 60%. However, in contrast, regions such as Somali and SNNP fell short in terms of retention rate, with only 41% and 42% of residents respectively retaining their VC.

## Model parameter results

The ICC offers a means to gauge the extent to which the discrepancy in the outcome across level-2 Units is accounted for. In this particular investigation, where children aged 12─23 months (i.e., the level-1 units) are arranged into clusters (i.e., the level-2 units), an ICC value of 0.38 derived from the null model would indicate that community-level factors accounted for 38% of the fluctuation in the odds of VC retention while the remaining 62% could be attributed to variances among study subjects.

Furthermore, the MOR of 3.7 conveys that, the median value of OR between clusters at a high rate of VC retention and clusters at the lowest retention rate when randomly choosing two children having the same individual-level characteristics but from a different cluster, the increased rate of VC retention when shifting from low risk to high risk is 3.7 times. The value of MOR and ICC indicates a justifiable reason for conducting multilevel logistic analysis (Table 2).

Proportional Change in Variance (PCV) or Change in Community-level Variance value of the full model (model III) indicates that about 8% of the variance in the odds of VC holding across the community/ class was attributed to the effect of both level -1 and level-2 factors.

From the table presented below, it is evident that the optimal model for the data at hand is the full model, denoted as model III, which includes both individual and community level factors. This conclusion is drawn based on the fact that model III exhibits a significantly lower value for deviance, BIC, and AICc, thus indicating its superior fit for the data.

Factors associated with VC retention following the multilevel multivariable regression analysis, it was observed that immunization status, place of residency, and community-level poverty emerged as significant factors associated with the retention rate of VC. According to the estimates derived from multivariable multilevel logistic analysis, the retention rate of VC is 65% lower among rural children (95% CI: 0.19─0.65) compared to those living in urban slums. Likewise, fully vaccinated children exhibited a 37% lower likelihood of retaining their VC (95% CI: 0.46─0.91) compared to those who were only partially vaccinated. Furthermore, a child born into a community with a higher poverty level has a 59% higher chance of retaining their vaccinations compared to a child born into a community with lower poverty levels (95% CI: 1.11─2.50) (Table 3).

**Table 1. Characteristics of the study participants in Ethiopia, 2016 (n = 1304).**

| Characteristics | Weighted frequency | Percent |
|---|---|---|
| **Maternal age** | | |
| 15—24 years | 316 | 24.27% |
| 25—34 years | 714 | 54.74% |
| 35—49 years | 274 | 20.99% |
| **Maternal education** | | |
| None | 737 | 56.54% |
| Primary school | 419 | 32.17% |
| Secondary and above | 147 | 11.29% |
| **Maternal Working status** | | |
| Working | 660 | 50.64% |
| not working | 644 | 49.36% |
| **Media access** | | |
| No access | 789 | 60.50% |
| Access | 515 | 39.50% |
| **Vaccination status** | | |
| Partial | 610 | 46.81% |
| Fully | 693 | 48.28% |
| **Sex of the child** | | |
| Male | 617 | 47.30% |
| Female | 687 | 52.70% |
| **Place of residency** | | |
| Urban | 207 | 15.88% |
| Rural | 1097 | 84.12% |
| **Region** | | |
| Tigray | 138 | 10.56% |
| Afar | 6 | 0.49% |
| Amhara | 264 | 20.22% |
| Oromia | 492 | 37.76% |
| Somali | 39 | 2.96% |
| Benishangul | 15 | 1.14% |
| SNNP | 283 | 21.73% |
| Gambela | 4 | 0.28% |
| Harari | 3 | 0.26% |
| Addis Ababa | 52 | 3.97% |
| Dire Dawa | 8 | 0.64% |
| **Community poverty** | | |
| Low | 683 | 52.44% |
| High | 620 | 47.56% |
| **Community literacy** | | |
| Low | 284 | 21.78% |
| High | 1019 | 78.22% |

## Discussion

This study aimed to investigate the rate of VC, as well as individual and community level variables associated with VC retention in Ethiopia. Multilevel logistic regression analysis was calibrated to identify factors associated with VC retention. In this study of caregivers, the retention rate for EPI cards was found to be 52.5%. This retention rate is lower than research

**Table 2. Model comparison and fitness parameter output.**

| Fitness parameter | Null Model | Model I | Model II | Model III |
|---|---|---|---|---|
| Community level variance | 1.89[95% CI: 1.21, 2.98] | 1.76 [95% CI: 1.08, 2.86] | 1.62[95% CI: 1.02, 2.60] | 1.74[95% CI: 1.06, 2.83] |
| Community level variance(se) | 0.4379891 | 0.436824 | 0.3913056 | 0.4340136 |
| ICC | 36.7% | 34.8% | 33% | 34.6% |
| MOR | 3.70 [95% CI: 2.84, 5.15] | 3.52 [95% CI: 2.70, 5.02] | 3.35[95% CI: 2.61, 4.63] | 3.50[95% CI: 2.66, 4.94] |
| PCV (%) | baseline | 7% | 14% | 8% |
| **Model fitness** | | | | |
| Log- likelihood ratio (LLR) | -775 | -742 | -753 | -740 |
| DIC(-2LLR) | 1500 | 1484 | 1506 | 1480 |
| AIC | 1554 | 1504 | 1515 | 1476 |
| BIC | 1564 | 1555 | 1535 | 1503 |

**Table 3. Multivariable multilevel analysis result of VC retention among children of 12–23 months in Ethiopia, 2016.**

| Characteristics | null model (95%CI AOR) | Model I (95%CI AOR) | Model II (95%CI AOR) | Model III (95%CI AOR) |
|---|---|---|---|---|
| Age group | | | | |
| 15—24 | | Ref | | Ref |
| 25—34 | | 1.28 [0.89, 1.84] | | 1.26[0.88, 1.82] |
| 35—49 | | 1.46 [0.91, 2.34] | | 1.42[0.89, 2.25] |
| **Educational status** | | | | |
| None | | Ref | | Ref |
| Primary | | 0.91 [0.63, 1.34] | | 0.90[0.62, 1.31] |
| Secondary& above | | 1.28 [0.75, 2.20] | | 1.15[0.62, 2.12] |
| **Working status** | | | | |
| Not working | | Ref | | Ref |
| Working | | 1.73 [1.43, 2.08] | | 1.05[0.76, 1.45] |
| **Media Access** | | | | |
| No | | Ref | | Ref |
| Yes | | 1.05 [0.76, 1.45] | | 1.34[0.93, 1.94] |
| **Immunization status** | | | | |
| Partially vaccinated | | **Ref** | | **Ref** |
| Fully vaccinated | | 0.66 [0.47, 0.93] | | **0.65[0.46, 0.91]** ** |
| **Place of residency** | | | | |
| Urban | | **Ref** | | **Ref** |
| Rural | | 0.25 [0.15, 0.44] | | **0.35[0.19, 0.65]** *** |
| | | **Community level factors** | | |
| **Community poverty** | | | | |
| High poverty | | | **Ref** | **Ref** |
| Low poverty | | | 0.42 [0.28, 0.63] | **1.59[1.11, 2.50]** ** |
| **Community literacy** | | | | |
| Low | | | **Ref** | **Ref** |
| High | | | 0.50[0.32, 0.78] | 0.82 [0.46, 1.48] |

Note

* = p-value 0.049–0.01

** = p-value <0.001, and

*** = p-value <0.0001

done in Nepal [21], India [22], and Nepal [23] which documented retention rates of 74%, 97.9% and 88.9% respectively. And it is higher than research done in Pakistan with a 33% retention rate [24]. The differences in vaccination coverage rates across these studies may be attributed to several factors, including variations in sample size, methodological approaches (e.g., survey, interview), and the level of information and awareness among caregivers in the respective countries. Using multilevel multivariable logistic regression analysis, we explored the associations between various factors and the retention of vaccination cards (VC). Our findings revealed that several factors were significantly associated with VC retention. Specifically, place of residency, immunization level and community level poverty.

This investigation has revealed that residing in a rural area is a mitigating factor affecting the retention rate of VC. This could be attributed to the likelihood that caregivers in rural slums may not fully comprehend the significance of VC, given their comparatively lower levels of education compared to urban residents. Additionally, it is conceivable that women living in rural areas of Ethiopia may not have access to safe and secure retrieval and storage systems for vaccination cards, in contrast to their urban counterparts. This discrepancy could lead to misplacement or damage to the cards.

The results of our analysis showed that children living in rural areas had lower vaccination coverage compared to their urban counterparts. This can be attributed to the challenges faced by rural communities in accessing healthcare services and vaccination programs. policymakers may have a vested interest in instituting interventions predicated on place of residency and vaccination status to foster the appropriate documentation of health cards generally, and VC. However, our findings also indicated that children from communities with lower levels of poverty had higher VC rate compared to children from high poverty. This suggests that socioeconomic status at the community level may have a stronger influence on vaccination uptake than the rural-urban divide. This finding suggests the importance of providing ongoing support to communities who have experienced financial scarcity to promote good documentation practices.

The completion of routine vaccinations during childhood has been identified as a significant factor contributing to the decrease in VC possession rate. The present discovery contradicts the research conducted in Nepal [23] which concluded that the completion of one's measles vaccine status has been discovered to be a significant contributing factor towards the increased holding of VC. Another study done in Uganda reported that children who possess VC are significantly more likely to receive all vaccinations, by a factor of ten than their peers who do not possess such cards [25]. This difference might be explained by the notion that parents of children who have completed their vaccination schedule may view the vaccination card as unnecessary, unlike families whose children are still undergoing vaccination courses. Another finding is low community level poverty. A child born from a community with low poverty level is likely to have retained VC compared to a child from a high poverty level. This finding is consistent with the study done in Cameroon [26].

While this inquiry has offered an exploratory view into the reporting of essential matters concerning child health, particularly regarding immunization cards, it is not without its limitations. Given that the study is cross-sectional, the relationships identified by the study are not going to be interpreted as causal factors. Another limitation of this study could be the insufficient inclusion of socio-economic variables. Furthermore, the study may be limited by its insufficient inclusion of socio-economic variables; incorporating a broader range of such variables could lead to more robust findings. The Negative correlation between completed vaccine status and card retention might not explained well and the reason is not known. So further investigation is required to bring satisfactory reason.

## Conclusion

The low retention rate of vaccination cards (VC) in Ethiopia is concerning, particularly since all caregivers are expected to retain their child's VC. Place of residency, wealth status, and vaccination status were factors that contributed to the change in the VC retention rate. Implement targeted campaigns to raise awareness about the importance of vaccination cards and the benefits of retaining them. Develop tailored interventions for populations identified as having lower VC retention rates, such as those residing in rural areas or with lower wealth status.

## Author Contributions

**Conceptualization:** Abiyu Abadi Tareke, Thomas Kidanemariam Yewodiaw.

**Data curation:** Abiyu Abadi Tareke.

**Formal analysis:** Abiyu Abadi Tareke, Enyew Woretaw Shiferaw, Habitu Birhan Eshetu.

**Investigation:** Abiyu Abadi Tareke.

**Methodology:** Abiyu Abadi Tareke.

**Software:** Abiyu Abadi Tareke, Habitu Birhan Eshetu.

**Supervision:** Abiyu Abadi Tareke, Ermias Bekele Enyew.

**Validation:** Abiyu Abadi Tareke.

**Visualization:** Abiyu Abadi Tareke.

**Writing – original draft:** Abiyu Abadi Tareke, Atikaw Tewabe Ayelign, Thomas Kidanemariam Yewodiaw, Habitu Birhan Eshetu.

**Writing – review & editing:** Abiyu Abadi Tareke, Atikaw Tewabe Ayelign, Thomas Kidanemariam Yewodiaw, Enyew Woretaw Shiferaw, Habitu Birhan Eshetu, Ermias Bekele Enyew.

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
