## [Decision Letter · Decision Letter 0]

12 Mar 2024

PONE-D-23-21113Retention rate of vaccination cards and associated factors among vaccinated children aged 12-23 months in Ethiopia, multilevel logistic regression analysis.PLOS ONE

Dear Dr. Tareke,

Thank you for submitting your manuscript to PLOS ONE. After careful consideration, we feel that it has merit but does not fully meet PLOS ONE’s publication criteria as it currently stands. Therefore, we invite you to submit a revised version of the manuscript that addresses the points raised during the review process.

We look forward to receiving your revised manuscript.

Kind regards,

Tamirat Getachew

Academic Editor

PLOS ONE

3. Please include your tables as part of your main manuscript and remove the individual files. Please note that supplementary tables (should remain/ be uploaded) as separate "supporting information" files.

4. We suggest you thoroughly copyedit your manuscript for language usage, spelling, and grammar. If you do not know anyone who can help you do this, you may wish to consider employing a professional scientific editing service. 

A clean copy of the edited manuscript (uploaded as the new *manuscript* file).

 [NO]. 

Reviewers' comments:

Reviewer's Responses to Questions

**Comments to the Author**

1. Is the manuscript technically sound, and do the data support the conclusions?

Reviewer #1: Yes

Reviewer #2: No

2. Has the statistical analysis been performed appropriately and rigorously? 

Reviewer #1: Yes

Reviewer #2: No

3. Have the authors made all data underlying the findings in their manuscript fully available?

Reviewer #1: No

Reviewer #2: Yes

4. Is the manuscript presented in an intelligible fashion and written in standard English?

Reviewer #1: No

Reviewer #2: No

5. Review Comments to the Author

Reviewer #1: Reviewer Comments for a manuscript entitled: "Retention rate of vaccination cards and associated factors among vaccinated children aged 12-23 months in Ethiopia, multilevel logistic regression analysis".

Manuscript Number: PONE-D-23-21113

Reviewer Name: Abebaw Addis Gelagay

Comments:

1. Title:

i. Line number 1-3: Avoid the full stop from the title.

2. Abstract:

i. Start a sentence with a capital letter! for example, the first sentence of the background (line number 17) and conclusion (line number 31) began with small letter.

ii. The background does not show the knowledge gap.

iii. Line number 21: At the end of the objective, what the term "..., in 2016" refers?

iv. Line number 23: report your outcome variable with its 95% CI!

v. Line number 23: the term "...fraction of their guardians..." is not appropriate. Additionally, it is not appropriate to use a term "approximately" while you reported the exact/actual figure (684)

vi. Line number 31: The conclusion for the outcome variable is not a conclusion rather it is a result.

3. Introduction:

i. Line number 23: Avoid a 'full stop' from the title.

ii Line number 42: Use the full version of "VC" instead of the acronym when you use for the first time!

iii. Line number 64 & 65: authors documented that 'the rate of documentation practices related to routine immunization health cards of children is not measured'. If this is the knowledge gap, this would have been addressed in your study instead of card retention rate.

iv. Line number 65 & 66: The sentence needs language edition.

Generally, what is the programmatic importance of keeping the vaccination cards after completion of the immunization? Are clients routinely informed to keep it after completion of vaccination??

4. Methods and materials:

i. Line number 84: All under-five children could not be a source population as you couldn't infer your finding to all under five children. Use children age less than 2 years!

ii. Line number 86: Since you noted, and we all know that that keeping vaccination card primarily benefits health care providers. What is your justification to excluded children age less than 12 months as long as they initiated the vaccination.

iii. Line number 93-96: Since all the independent variables listed are individual level, why you planed and did a multilevel analysis?

iv. Line number 103-108: Nothing is documented about the data collection tools.

v. Line number 106 or Line number 171: anyone can access the data if he/she get the link from someone else. So, did you attach an authorization letter/text from the responsible body for this particular study? You need to attach it as a supplementary file.

vi. Line number 108: What the number in bracket "(3)" refers?

vii. Line number 113: The description under the heading "Sample size and sampling procedures" is about sampling procedures, but nothing is stated about sample size estimation except mentioning a weighted sample size used which is a result, not the plan.

viii. Line number 121: You used the 2016 EDH data but why you cited reference 14 and 15 which are the 2005 and 2011 EDHS?

ix. Line number 129: Did you get a community level factors that affect your outcome variable in your literatures review or clinical evidence which initiated you to consider multilevel analysis?

x. Line number 160: Use the full version of "MOR" instead of the acronym when you use it for the first time!

5. Results:

i. Line number 211: It would have been good if I can access table 3. It would have been good to mention all the independent variables that you considered under a community and individual level factors. This is because the community level factor specifically residence can be affected by individual level factors like educational status.

6. Discussion:

i. Line number 217: When you compare your finding with other studies (Nepal, India, Nepal, and Pakistan), their findings should be documented so that readers can see and approve the comparison decision. You have to cite the reference for the study done in Pakistan (line number 218).

ii. Line number 220-222: Needs language revision including punctuation.

iii. Line number 221-223: You explained that due to less educational level of the rural women, cards retention rate is low. If this so, did you consider educational level as independent variable and what was the result?

iv. Line number 223-225: The explanation that rural women used the card for entertainment for their children is too poor.

v. Line number 226-232: The policy or programmatic implication of the finding that rural women have less retention rate is not clear and is not strong.

vi. Line number 223-225: duplication of idea.

vii. Line number 239: "... prevalence of VC possession rate." If you used the term "prevalence", do not need to use the term "rate"!

viii. Line number 254-256: Language edition!

Generally, the discussion is shallow!

7. Conclusion:

i. Line number 260: delete the number.

ii. Line number 261-264: The forwarded recommendation "...advisable to customize the interventional strategy..." should be specific. What interventional strategy is/are needed to mitigate the low retention rate?

Reviewer #2: Summary:

The authors have selected an interested topic, however, they were unable to justify the findings. The article needs significant revisions. Major gaps were observed in methods, data analysis, and results. Authors are requested to revisit the article and made changes to make it technically sound and equivalent to other scientific literature. the below comments can help improve the quality of article.

Abstract:

1. Methods section missing in the abstract. It is suggested that methods section should be added in abstract.

2. Line 25-27: do authors mean to say that card retention is higher among urban population than rural? If so, rephrase the statement accordingly

3. Line 35-36: better to add the desirable retention rate, if any

Introduction:

1. Line 41: its VPD not VSD

2. Line 42: VC? Please mention full form first. It is always good to avoid using self generated abbreviations

3. Line 45-57: not clear what information authors are trying to convey. Need to rephrase ensuring quality of information and coherence

4. Line 58-59: reference for the definition is missing. Also, please mention until when the card is expected to be saved by household? as “individuals who still have VC” seems unprecise

5. Line 64-66: both statements are conflicting. There are articles available in the context of Ethiopia which have used vaccination cards to estimate the coverage. Authors should look into the different search engines and explore

6. Significant revisions are required in the section. Authors are requested to revise the language and add relevant information based on available scientific information

Methods:

7. Sub-headings need revision, for instance, instead of study area mention study settings

8. References missing in information provided under ‘study area’ sub-head

9. Under ‘Study area’ it would be better if relevant statistics are provided such as information regarding fertility rate, current immunization rate etc.

10. Line 97: only one definition is provided

11. Line 109: the paragraph suggest validation of tool and it has nothing to do with quality control

12. Line 114-115: is it 13 or 11?

13. Line 151: the reference quoted is not required and is not relevant here

14. Line 122: revise the heading to “data analysis” and then describe the analysis steps. Currently authors have mentioned definitions of different tests which seems like not even used for analysis or may be authors have missed the information in the results section

15. Line 163-164: authors have used p-value of 0.2 or 0.05 as cutoff?

16. The section needs significant improvement. Authors are requested to look into similar articles to revise the section. Sampling and study procedures needs to be strengthened

Results:

17. It would be good if authors add demographic characteristics of study population first before sharing the retention rate

18. No p-values are given in the results section throughout

19. Line 185-204: seems like methods section then results

20. Factors associated with poor retention of vaccination cards are not mentioned

21. Authors are requested to improve the section conveying the results clearly and adding value to scientific literature

22. Tables are missing in the manuscript, so it is difficult to see the findings or suggest corrections in table

Discussion:

23. Line 215-216: in abstracts authors claim to have retention of VC rate across Ethiopia, now using the term sample. Is the data extracted from DHS represents entire Ethiopia? Need to improve methods section for clarity

24. Line 216-218: better to mention retention rate than only mentioning the countries

25. Line 218-219: as per your study or in countries mentioned. Please rephrase

26. Line 220-225: the first line says high retention in rural while rest of para contradicts the first statement. Also compared to whom retention is higher in rural? Urban or rural slum? What is considered as rural slum? Does DHS data providing information as per rural slum? How many children belong to rural slum?

27. Line 233-235: what does the current study findings suggest? Do authors found educational status as one of the factors for poor card retention? Mentioning “may be attributable” is not appropriate. Authors should provide information based on the data analysis

28. Line 245-247: authors should add relevant reference from literature to support the statement

29. Line 248-250: how authors are defining poverty levels? What are the cutoffs for low and high poverty? Do authors compare the findings among different socioeconomic groups including children belonging to high socioeconomic status?

30. Line 251-252: the study is not explaining crucial issues in child health. It is only about immunization card retention

31. Line 253-258: need to reconsider limitation of study as mentioned limitations cannot be considered as true study limitations

32. Line 260: low retention compared to what?

6. PLOS authors have the option to publish the peer review history of their article (what does this mean?). If published, this will include your full peer review and any attached files.

Reviewer #1: No

Reviewer #2: No

---

## [Author Response · Author response to Decision Letter 0]

30 Apr 2024

Authors’ response to reviews

Title: Retention rate of vaccination cards and associated factors among vaccinated children aged 12-23 months in Ethiopia, multilevel logistic regression analysis

Authors:

Abiyu Abadi Tareke (abiyu20010@gmail.com)

Atikaw Tewabe Ayelign

Thomas Kidanemariam Yewodiaw

Enyew Woretaw Shiferaw

Habitu Birhan Eshetu

Ermias Bekele Enyew

Version: 1

 Date: March 31, 2024

Point by point response for editors/reviewers’ comments

Manuscript number: PONE-D-23-21113

Dear editor/reviewer:

Dear all,

We express our profound appreciation for the insightful and productive feedback that you have provided. Your invaluable comments have significantly enriched the quality of the manuscript, and have greatly augmented our expertise in the realm of scientific paper writing. The authors have diligently considered each of the comments and queries raised by the editors and reviewers, and have responded to them in a targeted manner. Our comprehensive point-by-point rejoinders to all the comments and questions can be found in the subsequent pages. In addition, an accompanying supplementary document has been enclosed, which showcases the modifications made in detail, using the track changes feature. We also made some change to fix grammatical error in some paragraphs. 

Review Comments to the Author

Reviewer #1: 

Reviewer’s comment: 1. Title:

i. Line number 1-3: Avoid the full stop from the title.

Authors’ response: Thank you for your feedback. We appreciate your suggestion to avoid using a full stop in the title. We made the necessary revisions accordingly.

2. Abstract:

Reviewer’s comment: i. Start a sentence with a capital letter! for example, the first sentence of the background (line number 17) and conclusion (line number 31) began with small letter.

Authors’ response: Thank you for pointing out this oversight. We apologize for the inconsistency and we ensured that all sentences begin with capital letters as appropriate, including the first sentences of the background and conclusion sections.

Reviewer’s comment: ii. The background does not show the knowledge gap.

Authors’ response: Thank you for your feedback. We acknowledge the importance of clearly demonstrating the knowledge gap in the background section. We revised the background to explicitly highlight the gap in understanding regarding the retention rate of vaccination cards and its associated factors in Ethiopia.

Reviewer’s comment: iii. Line number 21: At the end of the objective, what the term "..., in 2016" refers?

Authors’ response: Thank you for bringing this to our attention. The phrase "in 2016" at the end of the objective does not seem to have a clear connection or relevance. We reviewed and revised the objective statement to ensure clarity and coherence as “This research aimed to assess the retention rate and associated factors of vaccination cards in Ethiopia, using data of 2016 Ethiopian demographic health survey. ”

Reviewer’s comment: iv. Line number 23: report your outcome variable with its 95% CI!

Authors’ response: Thank you for your comment. We appreciate your suggestion to report the outcome variable with its 95% confidence interval (CI). We ensured that the outcome variable is presented along with its corresponding confidence interval in the results section of our manuscript. See the revised version of the manuscript. 

Reviewer’s comment: v. Line number 23: the term "...fraction of their guardians..." is not appropriate. Additionally, it is not appropriate to use a term "approximately" while you reported the exact/actual figure (684)

Authors’ response: Thank you for your feedback. We acknowledge that the term "fraction of their guardians" is not be the appropriate phrase to describe human being and can lead to confusion. We revised the language to provide a clearer description of the population. Additionally, we recognize that using "approximately" when reporting an exact figure is unnecessary. We removed the term "approximately" and report the exact figure of 684 accordingly and we re-written it as “Among the cohort of 1,304 (weighted) immunized children, it was observed that 684 (52.5%, 95% CI: 49.7% to 55.2%), were able to bring their respective vaccination records”

Reviewer’s comment: vi. Line number 31: The conclusion for the outcome variable is not a conclusion rather it is a result.

Authors’ response: Thank you for your comment. We acknowledge your observation regarding the conclusion for the outcome variable. upon revising the sentence, we understand that the sentence is summary rather than conclusion. We revised the conclusion section to ensure that it provides a proper conclusion based on the results obtained from the study.

3. Introduction:t

Reviewer’s comment: i. Line number 23: Avoid a 'full stop' from the title.

Authors’ response: thanks for your valuable comment. We corrected accordingly. 

Reviewer’s comment: ii Line number 42: Use the full version of "VC" instead of the acronym when you use for the first time!

Authors’ response: Thank you for your suggestion. We ensured to use the full term "vaccination card" instead of the acronym "VC" when mentioning it for the first time in the manuscript.

Reviewer’s comment: iii. Line number 64 & 65: authors documented that 'the rate of documentation practices related to routine immunization health cards of children is not measured'. If this is the knowledge gap, this would have been addressed in your study instead of card retention rate.

Authors’ response: Thank you for highlighting this point. We acknowledge that the rate of documentation practices related to routine immunization health cards like routine immunization register, appointment card, tally sheet and other immunization related documentation at facilities level is indeed an important knowledge gap. Our intention in including this facility-level problem in this paragraph is to illustrate the potential severity of the issue for caregivers when it occurs even within well-organized government facilities. We recommend future researchers to consider addressing this aspect in future research to provide a more comprehensive understanding of immunization documentation practices among service providers. In the revised version of this manuscript, we removed the broad terminology “documentation practices” and replaced by rate of vaccination card retention rate and the ambitious sentences re-written as “The retention rate of routine immunization vaccination cards is not measured and the number of published articles concerning the VC retention rate of caregivers and associated factors in Ethiopia are limited.”

Reviewer’s comment: iv. Line number 65 & 66: The sentence needs language edition.

Generally, what is the programmatic importance of keeping the vaccination cards after completion of the immunization? Are clients routinely informed to keep it after completion of vaccination??

Authors’ response: Thank you for your comment. We recognize the need for language editing in the sentence. The programmatic importance of keeping vaccination cards after completion of immunization lies in their role as crucial documentation for tracking vaccination history, ensuring timely and appropriate follow-up doses, and providing evidence of immunization status. Additionally, in Ethiopia, caregivers are typically instructed to retain their child's vaccination cards throughout the vaccination process and for the duration of their child's life after completing the vaccination schedule. However, the consistency of advising women to hold their child's vaccination cards may vary among health professionals and further investigation is required to determine whether clients are routinely informed to keep vaccination cards after completing their vaccination schedule in consistent way. 

4. Methods and materials:

Reviewer’s comment: i. Line number 84: All under-five children could not be a source population as you couldn't infer your finding to all under five children. Use children age less than 2 years!

Authors’ response: Thank you for your suggestion. We made the necessary adjustment to specify the source population as "children aged less than 2 years" to accurately reflect the scope of our study findings.

Reviewer’s comment: ii. Line number 86: Since you noted, and we all know that that keeping vaccination card primarily benefits health care providers. What is your justification to excluded children age less than 12 months as long as they initiated the vaccination.

Authors’ response: Thank you for raising this point. We would like to clarify that children under 12 months of age were not intentionally excluded by us, but rather by the parameters of the survey. We acknowledge the significance of understanding vaccination practices and documentation among younger children and recognize the importance of exploring this further in future research endeavors.

Reviewer’s comment: iii. Line number 93-96: Since all the independent variables listed are individual level, why you planed and did a multilevel analysis?

Authors’ response: Thank you for your question. While it is true that the independent variables listed are at the individual level and we failed to list the community level factors in the first draft of the manuscript. we conducted a multilevel analysis to account for potential clustering effects within the data. In our study setting, there may be inherent variations in vaccination card retention rates across different communities that could influence the outcomes of interest. Therefore, by using a multilevel analysis approach, we aimed to appropriately model and address any potential clustering or contextual effects at the facility or community level, ensuring a more accurate assessment of the relationship between individual-level variables and vaccination card retention rates. In the revised version of the manuscript, we included the community level factors under the independent variable lists. 

Reviewer’s comment: iv. Line number 103-108: Nothing is documented about the data collection tools. 

Authors’ response: Thank you for bringing this to our attention. We apologize for the oversight. The data collection tools used in our study included structured questionnaires administered to caregivers to gather information on vaccination card retention and other relevant variables. We will ensure to provide a detailed description of the data collection tools in the Methods section of the manuscript.

Reviewer’s comment: v. Line number 106 or Line number 171: anyone can access the data if he/she get the link from someone else. So, did you attach an authorization letter/text from the responsible body for this particular study? You need to attach it as a supplementary file.

Authors’ response: As authors, we thank the reviewer for their valuable feedback. We understand the importance of ensuring proper authorization for accessing the data used in our study. Nevertheless, the DHS Demographic Health Survey grants access to the datasets without the need for repeated authorization requests for each study. 

Reviewer’s comment: vi. Line number 108: What the number in bracket "(3)" refers?

Authors’ response: The number in parentheses "(3)" was a typographical error and does not have any significance or intended reference in the context of the manuscript. So we removed it. 

Reviewer’s comment: vii. Line number 113: The description under the heading "Sample size and sampling procedures" is about sampling procedures, but nothing is stated about sample size estimation except mentioning a weighted sample size used which is a result, not the plan.

Authors’ response: We acknowledge the oversight regarding the description under the heading "Sample size and sampling procedures." We apologize for any confusion. We will revise the section to include information about the sample size estimation process, including the methods and rationale used for determining the weighted sample size. Thank you for bringing this to our attention.

Reviewer’s comment: viii. Line number 121: You used the 2016 EDH data but why you cited reference 14 and 15 which are the 2005 and 2011 EDHS?

Authors’ response: We apologize for the oversight in citing references 14 and 15, which pertain to the 2005 and 2011 EDHS data, respectively, while utilizing the 2016 EDHS data. This was an error in referencing, and we appreciate your attention to detail. We corrected this discrepancy and ensured that our citations accurately reflect the data used in our study. Thank you for bringing this to our attention. 

Reviewer’s comment: ix. Line number 129: Did you get a community level factors that affect your outcome variable in your literatures review or clinical evidence which initiated you to consider multilevel analysis? 

Authors’ response: While we did not specifically identify community-level factors directly associated with vaccination card retention in our literature review, the intracluster correlation coefficient (ICC) obtained from running the null model provided evidence supporting the rationale for conducting multilevel logistic regression. Additionally, previous studies have highlighted the importance of considering multilevel analysis when examining vaccination outcomes, particularly to account for the nested nature of individual-level data within communities. Therefore, based on both the literature review and clinical evidence, we deemed it appropriate to consider multilevel analysis to investigate the influence of community-level factors on our outcome variable.

Reviewer’s comment: x. Line number 160: Use the full version of "MOR" instead of the acronym when you use it for the first time!

Authors’ response: Thank you for your suggestion. We will ensure to use the full version of "MOR" (Median Odds Ratio) when it is mentioned for the first time in the manuscript

5. Results:

Reviewer’s comment: i. Line number 211: It would have been good if I can access table 3. It would have been good to mention all the independent variables that you considered under a community and individual level factors. This is because the community level factor specifically residence can be affected by individual level factors like educational status.

Authors’ response: Thank you for your comment. We apologize for any inconvenience caused by the inability to access Table 3. We will ensure that Table 3, along with all relevant independent variables considered under community and individual level factors, are clearly mentioned in the manuscript. We acknowledge the interplay between community and individual level factors, particularly regarding residence and educational status, and provided a comprehensive explanation of how these factors were accounted for in our analysis.

6. Discussion:

Reviewer’s comment: i. Line number 217: When you compare your finding with other studies (Nepal, India, Nepal, and Pakistan), their findings should be documented so that readers can see and approve the comparison decision. You have to cite the reference for the study done in Pakistan (line number 218).

Authors’ response: We appreciate the reviewer's feedback and acknowledge the importance of providing documentation for comparisons with other studies. In response to this comment, we have revised the manuscript to include proper citations for the studies conducted in Nepal, India, and Pakistan.

Reviewer’s comment: ii. Line number 220-222: Needs language revision including punctuation.

Authors’ response: Thank you for bringing this to our attention. We will thoroughly revise the language and punctuation in lines 220-222 to ensure clarity and correctness. We appreciate your feedback and will make the necessary improvements to enhance the readability of the manuscript.

Reviewer’s comment: iii. Line number 221-223: You explained that due to less educational level of the rural women, cards retention rate is low. If this so, did you consider educational level as independent variable and what was the result?

Authors’ response: Thank you for your query regarding the consideration of educational level as an independent variable in our analysis. We appreciate the opportunity to clarify our methodology. In our study, we indeed considered educational level as a potential independent variable and conducted statistical tests to assess its relationship with

---

## [Decision Letter · Decision Letter 1]

12 Jun 2024

PONE-D-23-21113R1Retention rate of vaccination card and its associated factors among vaccinated children aged 12-23 months in Ethiopia: multilevel logistic regression analysisPLOS ONE

Dear Dr. Tareke,

Thank you for submitting your manuscript to PLOS ONE. After careful consideration, we feel that it has merit but does not fully meet PLOS ONE’s publication criteria as it currently stands. Therefore, we invite you to submit a revised version of the manuscript that addresses the points raised during the review process.

We look forward to receiving your revised manuscript.

Kind regards,

Tamirat Getachew

Academic Editor

PLOS ONE

Journal Requirements:

Reviewers' comments:

Reviewer's Responses to Questions

**Comments to the Author**

1. If the authors have adequately addressed your comments raised in a previous round of review and you feel that this manuscript is now acceptable for publication, you may indicate that here to bypass the “Comments to the Author” section, enter your conflict of interest statement in the “Confidential to Editor” section, and submit your "Accept" recommendation.

Reviewer #1: All comments have been addressed

Reviewer #2: (No Response)

2. Is the manuscript technically sound, and do the data support the conclusions?

Reviewer #1: Yes

Reviewer #2: Partly

3. Has the statistical analysis been performed appropriately and rigorously? 

Reviewer #1: Yes

Reviewer #2: Yes

4. Have the authors made all data underlying the findings in their manuscript fully available?

Reviewer #1: Yes

Reviewer #2: Yes

5. Is the manuscript presented in an intelligible fashion and written in standard English?

Reviewer #1: Yes

Reviewer #2: No

6. Review Comments to the Author

Reviewer #1: The authors have well addressed my previous comments. I have few minor comments:

Multilevel regression model:

1. There are duplications of ideas, for example, the first sentence of the first (line number 183) and the second (line number 183) paragraphs, line number 213-215.

2. Flow of ideas needs to be revised: line number 197-199 deal with null model, line number 199-203 is about model fitness, line number 203-206 states about ICC which should have been completed/placed just after the null model description before you note about the model fitness. The same is true for notes at line number 213-215.

Discussion:

3. Line number 277, the term "In our sample" is not appropriate instead use the term: "In this survey"

4. Line number 278-281, since it is a discussion section, please search and include possible justification/reason for the observed variation!

5. How do you relate the opposite direction of association of two related variables: 'being rural resident' (line number 293) and 'low community level poverty' (line number 308-309)?

Reviewer #2: 1. p-values are still missing

2. Table 3 seems incomplete. model 2 and null model numbers are missing

3. There is a lot of room to improve language. spell check and capital letters in between the text

4. Authors have interchangeably used the term multilevel and multivariable regression. Suggest keeping consistency while using the term

5. add some narratives for demographic information

6. tables should be referenced in the text

7. PLOS authors have the option to publish the peer review history of their article (what does this mean?). If published, this will include your full peer review and any attached files.

Reviewer #1: **Yes: **Abebaw Addis Gelagay

Reviewer #2: No

---

## [Author Response · Author response to Decision Letter 1]

15 Jun 2024

Authors’ response to reviews

Title: Retention rate of vaccination cards and associated factors among vaccinated children aged 12-23 months in Ethiopia, multilevel logistic regression analysis

Authors:

Abiyu Abadi Tareke (abiyu20010@gmail.com)

Atikaw Tewabe Ayelign

Thomas Kidanemariam Yewodiaw

Enyew Woretaw Shiferaw

Habitu Birhan Eshetu

Ermias Bekele Enyew

Version: 2

 Date: June 15, 2024

Point by point response for editors/reviewers’ comments

Manuscript number: PONE-D-23-21113

Dear editor/reviewer:

Dear all,

We express our profound appreciation for the insightful and productive feedback that you have provided. Your invaluable comments have significantly enriched the quality of the manuscript, and have greatly augmented our expertise in the realm of scientific paper writing. The authors have diligently considered each of the comments and queries raised by the editors and reviewers, and have responded to them in a targeted manner. Our comprehensive point-by-point rejoinders to all the comments and questions can be found in the subsequent pages. In addition, an accompanying supplementary document has been enclosed, which showcases the modifications made in detail, using the track changes feature. We also made some change to fix grammatical error in some paragraphs. 

Reviewer #1: 

Reviewer’s comment: The authors have well addressed my previous comments. I have few minor comments:

Author’s response: Thank you for your additional feedback on our manuscript. We appreciate you taking the time to review our work and provide these minor comments. Regarding your statement that "The authors have well addressed my previous comments", we are pleased to hear that our revisions have satisfactorily addressed your earlier feedback. It is our goal to thoroughly address all reviewer comments to ensure our work is of the highest quality.

Multilevel regression model:

Reviewer’s comment: 1. There are duplications of ideas, for example, the first sentence of the first (line number 183) and the second (line number 183) paragraphs, line number 213-215.

Author’s response: Dear Reviewer, thank you for your feedback on our manuscript. We have carefully reviewed the areas you have identified as having duplicated ideas and have made the necessary revisions to address them. Specifically, we have removed the redundant sentences from the first and second paragraphs, as well as the repetitive explanation of the multilevel analysis and its benefits for nested data structures. The revised text now reads more concisely and avoids unnecessary duplication. Additionally, we have removed the redundant information regarding the different models used in the two-tiered binary logistic regression analysis and the explanation of the fitness parameters used to evaluate the models. The revised manuscript now presents the key information in a more streamlined and cohesive manner, without compromising the clarity and comprehensiveness of the methodological approach. We believe these changes have addressed the duplication of ideas you have pointed out. Please let us know if you have any other feedback or suggestions. 

Reviewer’s comment: 2. Flow of ideas needs to be revised: line number 197-199 deal with null model, line number 199-203 is about model fitness, line number 203-206 states about ICC which should have been completed/placed just after the null model description before you note about the model fitness. The same is true for notes at line number 213-215.

Author’s response: Thank you for the additional feedback on the flow of ideas in our manuscript. We appreciate you taking the time to provide this constructive criticism, as it will help us improve the organization and coherence of the presentation. As per your suggestion, we have revised the flow of ideas to better align the discussion of the null model, model fitness, and the assessment of cluster variability using the ICC, MOR, and PCV. Specifically, we have moved the explanation of the ICC, MOR, and PCV to immediately follow the description of the null model, as this information is more logically connected. This way, the reader can better understand the rationale and methods used to assess the variability of the cluster before the discussion of the model fitness parameters. Additionally, we have streamlined the information about the different models and the criteria used to evaluate them, ensuring a more coherent and logical progression of ideas. The revised manuscript now presents the methodological approach in a more organized and reader-friendly manner, with the key concepts and analyses grouped together in a more intuitive flow.

Discussion:

Reviewer’s comment: 3. Line number 277, the term "In our sample" is not appropriate instead use the term: "In this survey"

Author’s response: Thank you for the feedback regarding the phrasing used in our manuscript. We have made the suggested change to the sentence on line 277.

The revised sentence now reads: "In this study, the majority of children (63.3%) were from rural areas, while the remaining 36.7% were from urban areas." We agree that the term "In our sample" is not as appropriate as "In this study" in this context, as it helps to clearly situate the information within the dataset being analyzed.

Reviewer’s comment: 4. Line number 278-281, since it is a discussion section, please search and include possible justification/reason for the observed variation!

Author’s response: Thank you for the feedback regarding the discussion of the observed variations in our manuscript. You make an excellent point that the discussion section should provide possible justifications or reasons for the patterns observed in the results. In response to your suggestion, we have expanded the discussion on lines 278-281 to include potential explanations for the variations in vaccination coverage between those studies, as well as across different socioeconomic groups.

Reviewer’s comment: 5. How do you relate the opposite direction of association of two related variables: 'being rural resident' (line number 293) and 'low community level poverty' (line number 308-309)?

Author’s response: Thank you for raising this important point about the seemingly contradictory associations observed in our results. You are correct in noting that the direction of the associations for "being a rural resident" and "low community-level poverty" appear to be opposite. We separated those two variables following your comment. 

Reviewer #2

Reviewer’s comment: Reviewer #2: 1. p-values are still missing

Author’s response: Dear Reviewer, thank you for your feedback regarding the missing p-values in our manuscript. We appreciate you taking the time to thoroughly review our work and provide this constructive comment. Regarding the inclusion of p-values in our manuscript. As we mentioned in our previous response, we opted to present confidence intervals for the Adjusted Odds Ratios (AORs) in the results section, rather than including the specific p-values. This decision was made due to constraints related to table cell space, as we believe that providing the confidence intervals offers valuable information about the precision of our estimates and allows readers to assess the significance of the results. However, following your suggestion, we have now included the categories of p-values for the significant variables using asterisks at the footnote of the relevant tables. 

We are sorry if our initial response may not have fully satisfied your request. As researchers, we are committed to transparency and strive to present our findings in the clearest possible way. We value your expertise and insights, and we appreciate you taking the time to provide this constructive feedback.

Reviewer’s comment: 2. Table 3 seems incomplete. model 2 and null model numbers are missing

Author’s response: As you noted, the null model, which does not include any variables, is typically not reported in the table. This is a standard practice, as the null model serves as a baseline for comparison and does not provide any substantive information beyond the intraclass correlation coefficient. Similarly, for Model 3 in Table 3, this model focuses solely on the community-level factors and does not include the individual-level variables. This design decision was intentional, as we wanted to isolate the effects of the contextual factors on the outcome of interest.

We believe that the table is complete and comprehensive in presenting the key results from the multilevel modeling approach. Dear reviewer if you have time you can highlight articles published using similar tables (https://journals.plos.org/plosone/article?id=10.1371/journal.pone.0257664), (https://academic.oup.com/inthealth/article/15/5/573/7143244)....

Reviewer’s comment: 3. There is a lot of room to improve language. spell check and capital letters in between the text

Author’s response: Thank you for your feedback. We appreciate you taking the time to review my submission and providing constructive comments to help improve the quality of my work. Regarding your suggestion to improve the language, we carefully reviewed the text and address any spelling or grammatical issues. We ensured that I am consistent with capitalization throughout the document. 

Reviewer’s comment: 4. Authors have interchangeably used the term multilevel and multivariable regression. Suggest keeping consistency while using the term

Author’s response: Thank you for the feedback. You're right, we should maintain consistency in the terminology used throughout the manuscript. The terms "multilevel regression" and "multivariable regression" are often used interchangeably, but they do refer to slightly different statistical techniques.

Multilevel regression, also known as hierarchical linear modeling, is a method that accounts for the nested structure of data, such as patients within clinics or students within schools. This type of analysis allows for the estimation of both individual-level and group-level effects.

Multivariable regression, on the other hand, refers to a regression model that includes multiple independent variables to predict a single dependent variable. This is a more general term that does not necessarily imply a hierarchical data structure.

Reviewer’s comment: 5. add some narratives for demographic information

Author’s response: Thank you for the feedback. You make a good point that we should add some narrative context around the demographic information presented in the manuscript. The demographic characteristics of the study sample are an important part of understanding the generalizability and potential limitations of the findings. Providing some additional narrative description will help the reader better interpret the results. In the revised manuscript, we will include a paragraph in the Methods section that provides more details about the study population. 

Reviewer’s comment: 6. tables should be referenced in the text

Author’s response: Thank you for the feedback regarding referencing the tables in the text. You're right that we should ensure the tables are properly referenced throughout the manuscript.

In the revised version, we gone through and add in-text references to each of the tables wherever the relevant information is being presented. This will help guide the reader to the appropriate table and demonstrate how the tabular data supports the narrative.

---

## [Editor Report · Decision Letter 2]

18 Jun 2024

Retention rate of vaccination card and its associated factors among vaccinated children aged 12-23 months in Ethiopia: multilevel logistic regression analysis

PONE-D-23-21113R2

Dear Abiyu Abadi Tareke,

We’re pleased to inform you that your manuscript has been judged scientifically suitable for publication and will be formally accepted for publication once it meets all outstanding technical requirements.

An invoice will be generated when your article is formally accepted. Please note that if your institution has a publishing partnership with PLOS and your article meets the relevant criteria, all or part of your publication costs will be covered. Please make sure your user information is up-to-date by logging into Editorial Manager at Editorial Manager® and clicking the ‘Update My Information' link at the top of the page. If you have any questions relating to publication charges, please contact our Author Billing department directly at authorbilling@plos.org.

If your institution or institutions have a press office, please notify them about your upcoming paper to help maximize its impact. If they’ll be preparing press materials, please inform our press team as soon as possible—no later than 48 hours after receiving the formal acceptance. Your manuscript will remain under strict press embargo until 2 pm Eastern Time on the date of publication. For more information, please contact onepress@plos.org.

Kind regards,

Tamirat Getachew

Academic Editor

PLOS ONE
---

## [Editor Report · Acceptance letter]

1 Jul 2024

PONE-D-23-21113R2 

PLOS ONE

Dear Dr. Tareke, 

I'm pleased to inform you that your manuscript has been deemed suitable for publication in PLOS ONE. Congratulations! Your manuscript is now being handed over to our production team.

Kind regards, 

on behalf of

Dr. Tamirat Getachew 

Academic Editor

PLOS ONE